# Resource Utilization of Acid Mine Drainage (AMD): A Review

**Jiaqiao Yuan** [1], **Zhan Ding** [1], **Yunxiao Bi** [1], **Jie Li** [1], **Shuming Wen** [1,2,3] and **Shaojun Bai** [1,2,3,*]

1   Faculty of Land Resource Engineering, Kunming University of Science and Technology, Kunming 650093, China; yuankust@126.com (J.Y.); kustdingzhan@126.com (Z.D.); byxkust@126.com (Y.B.); lijiekust@126.com (J.L.); shmwen@126.com (S.W.)
2   State Key Laboratory of Complex Nonferrous Metal Resources Clean Utilization, Kunming University of Science and Technology, Kunming 650093, China
3   Yunnan Key Laboratory of Green Separation and Enrichment of Strategic Mineral Resources, Yunnan Provincial Department of Science and Technology, Kunming 650093, China
*   Correspondence: baishaojun830829@126.com; Tel.: +86-871-65187288

**Abstract:** Acid mine drainage (AMD) is a typical type of pollution originating from complex oxidation interactions that occur under ambient conditions in abandoned and active mines. AMD has high acidity and contains a high concentration of heavy metals and metalloids, posing a serious threat to ecological systems and human health. Over the years, great progress has been made in the prevention and treatment of AMD. Remediation approaches like chemical neutralization precipitation, ion exchange, membrane separation processes, and bioremediation have been extensively reported. Nevertheless, some limitations, such as low efficacy, excessive consumption of chemical reagents, and secondary contamination restrict the application of these technologies. The aim of this review was to provide updated information on the sustainable treatments that have been engaged in the published literature on the resource utilization of AMD. The recovery and reuse of valuable resources (e.g., clean water, sulfuric acid, and metal ions) from AMD can offset the cost of AMD remediation. Iron oxide particles recovered from AMD can be applied as adsorbents for the removal of pollutants from wastewater and for the fabrication of effective catalysts for heterogeneous Fenton reactions. The application of AMD in beneficiation fields, such as activating pyrite and chalcopyrite flotation, regulating pulp pH, and leaching copper-bearing waste rock, provides easy access to the innovative utilization of AMD. A review such as this will help researchers understand the progress in research, and identify the strengths and weaknesses of each treatment technology, which can help shape the direction of future research in this area.

**Keywords:** acid mine drainage (AMD); valuable resources; comprehensive utilization; Fenton oxygen processes; beneficiation

## 1. Introduction

Pollution occurs inevitably during the extraction of mineral resources, and acid mine drainage (AMD) is one of the most significant challenges facing the global mining industry. AMD is primarily generated by the oxidation of sulfide minerals when exposed to air, water, and microbial activity, and it is widespread in active or abandoned polymetallic sulfide and coal mines [1–5]. AMD is strongly acidic and the pH value is commonly below 4, which has generally high concentrations of sulfate and dissolved metal ions [6–8]. However, the discharge of untreated AMD has the risk of contaminating nearby water sources, flora, and fauna. This not only has serious negative impacts on biodiversity but also causes high levels of metal ions in the soil to reach the top of the food chain, ultimately creating a risk to human health [9–11]. Therefore, the treatment of AMD is one of the most important hot topics in the field of the environment.

Over the past decades, researchers have been making every endeavor toward AMD remediation by degrading acidity and removing metal ions to reduce the impact of AMD

on the environment and human health [12–14]. The available treatment methods confirmed that lime neutralization is a frequently applied and low-cost technique for AMD remediation. However, the main drawback of this method is the generation of a large volume of sludge, which is not conducive to the comprehensive utilization of valuable metal ions and sulfates from AMD [15,16]. Other conventional approaches such as ion exchange [17], chemical precipitation [18], adsorption [19], and membrane separation techniques [20,21] are also frequently employed to treat AMD [22]. However, most of these techniques mainly focus on the removal of acidity, metal, and sulfate content of the AMD itself and fail to recover and reuse the valuable components of AMD adequately [23].

Environmental protection and rehabilitation are fast becoming a priority. Therefore, assiduous efforts on the recovery of valuable AMD constituents (such as clean water, sulfuric acid, and metal ions) have been ongoing [24–27]. Some researchers have considered AMD as a potential low-cost raw material for the production of iron oxide particles (goethite, ferrihydrite, and magnetite) [28–30]. These iron oxides have been previously shown to be effective catalysts for heterogeneous Fenton reactions [31], and also as adsorbents for the removal of pollutants from wastewater [32,33]. Additionally, the removal of pollutants from industrial wastewaters can be achieved with the aid of AMD due to the fact that iron ions and hydrogen ions originating from AMD can promote the Fenton reaction over a wider pH range, producing hydroxyl radicals (•OH) with the strong oxidizing ability and can avoid the formation of iron sludge [34]. Accordingly, the recovery of valuable resources from AMD is a promising approach to offset the cost of AMD treatment.

Recently, the utilization of AMD as an activator for pyrite/chalcopyrite inhibition by lime has been proposed by researchers and explored in the laboratory [35]. The results showed that AMD activation flotation of lime-depressed pyrite is feasible with a favorable flotation index, which will provide a new path to the sustainable exploitation and clean production of copper sulfide ore. The application of AMD to beneficiation production is one of the most efficient source treatments to reduce AMD pollution. The new technologies and crafts on the AMD treatments have been studied extensively, however, at present, the corresponding latest research progress in the resource utilization of AMD fails to summarize in detail. With that in mind, the aim of this review is to critically analyze the broad-spectrum treatment methods that have been engaged in the published literature on the resource utilization of AMD. The comprehensive resource utilization and future prospects of AMD in the field of mineral processing are discussed with emphasis. A review such as this will help researchers understand the progress in research, and identify the strengths and weaknesses of each treatment technology, which can help shape the direction of future research in this area.

## 2. Formation and Hazard of Acid Mine Drainage

### 2.1. Formation of Acid Mine Drainage

AMD is one of the main pollutants causing environmental impacts and its main sources include leachate from abandoned mines and underground deposits, leach water from open pits, tailings pits and waste rock dumps, and wastewater discharged during ore flotation and smelting [36–38]. Generally, AMD is generated by the oxidation of sulfide minerals due to exposure to air, water, and microbial activity [39,40]. AMD formation and corresponding contamination pathways are shown in Figure 1 [26].

Pyrite is one of the most abundant and widespread sulfide minerals and is extensively considered to be the predominant cause of AMD generation [41–43]. Moreover, arsenopyrite, chalcopyrite, galena, pyrrhotite, and sphalerite also contribute to the formation of AMD [42,44–46]. The series of chemical reactions involved in AMD formation are presented in Figure 2 [47]. The pyrite exposed to water and air is oxidized firstly, releasing $H^+$, $SO_4^{2-}$, and $Fe^{2+}$ ions at the surface, and the $Fe^{2+}$ ions continue to be further oxidized to $Fe^{3+}$ in water exposed to air. At low pH, $Fe^{3+}$ ions will be hydrolyzed and precipitated to form $Fe(OH)_3$ precipitation, while part of the $Fe^{3+}$ ions may continue to oxidize pyrite to produce sulfate and acid, which leads to AMD formation [39,48,49].

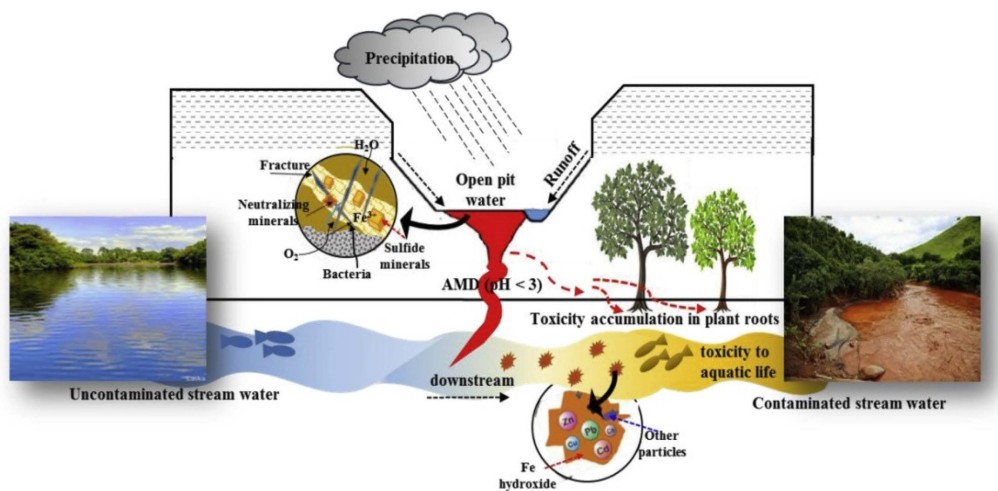

**Figure 1.** AMD formation and the corresponding contamination pathways. Adapted with permission from [26], copyrighter Elsevier, year 2019.

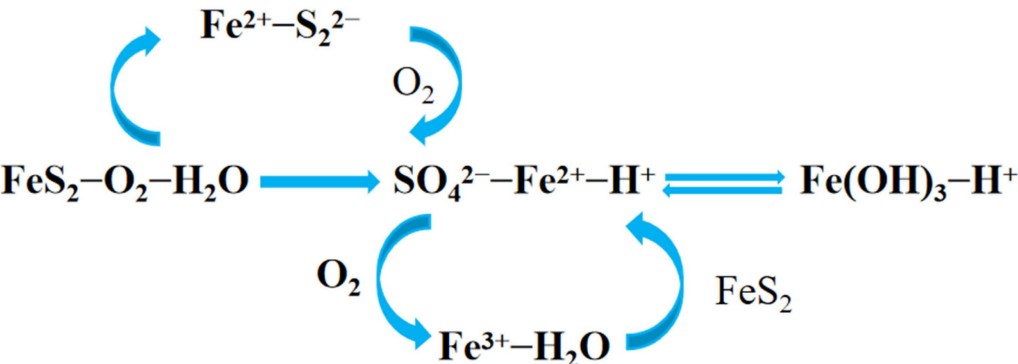

**Figure 2.** Chemical reactions related to AMD formation.

### *2.2. Hazards of Acid Mine Drainage*

　　Table 1 summarizes the chemical composition of AMD generated from typical mines across the world. From these data, it can be concluded that the AMD close to mining sites is generally highly acidic (pH = 2–4) and predominantly sulfate (1000–30,000 mg/L) and iron ion (100–2000 mg/L) [26]. Besides, the AMD contains major metals such as Cu, Zn, Mn, and Al. Cu is one of the most common heavy metals that are harmful to aquatic organisms. Although it is needed for their growth and reproduction, any small accumulation beyond the required amount can cause irreversible harm to some of these organisms. Thus, these findings indicate that the property of AMD discharged from mines in different locations varies greatly during mining and operation, resulting in different levels of hazard to the surrounding environment. AMD causes a range of environmental issues when it flows into groundwater, rivers, and lakes, and has serious implications for human health. Many researchers have conducted studies on the ecotoxicology of AMD. Some studies reported that AMD is toxic, as in Figure 3 [50].

**Table 1.** Chemical components of acid mine drainage from typical mines across the world.

| Country | Typical Mines | pH | Fe (mg/L) | Zn (mg/L) | Cu (mg/L) | Mn (mg/L) | Al (mg/L) | $SO_4^{2-}$ (mg/L) | References |
|---|---|---|---|---|---|---|---|---|---|
| China | Polymetallic Mine | 2.50 | 2490 | 500 | 2670 | 6590 | - | 24,530 | [51] |
| South Africa | Mpumalanga Coal Mine | 2 | 8000 | - | - | 75 | 300 | 30,000 | [52] |
| Spain | Riotinto Mine | 2.60–2.80 | 1824 | 557 | 184 | 329 | 2830 | 24,700 | [53] |
| Australia | Mount Morgan Gold Mine | 2.70 | 66 | 55 | 65 | 245 | 2317 | 29,547 | [54] |
| USA | Elizabeth Copper Mine | 3.30 | 123 | - | - | 2.60 | 13 | 1200 | [55] |
| Chile | Copper mine | 2.50 | 627.50 | - | 2298 | 224.50 | 1139 | 14,337 | [56] |
| Korea | Taejeong Coal Mine | 3.28 | 186 | - | - | 13 | 40 | 1950 | [57] |
| Canada | Gold Mine | 2.39 | 788 | 0.25 | 3.42 | 19.40 | 310 | 4520 | [58] |
| Brazil | Coal Mine | 2.33 | 611.38 | 62.65 | - | 37.98 | 269.37 | 7410.40 | [59] |

Notes: "-": Below the detection limits.

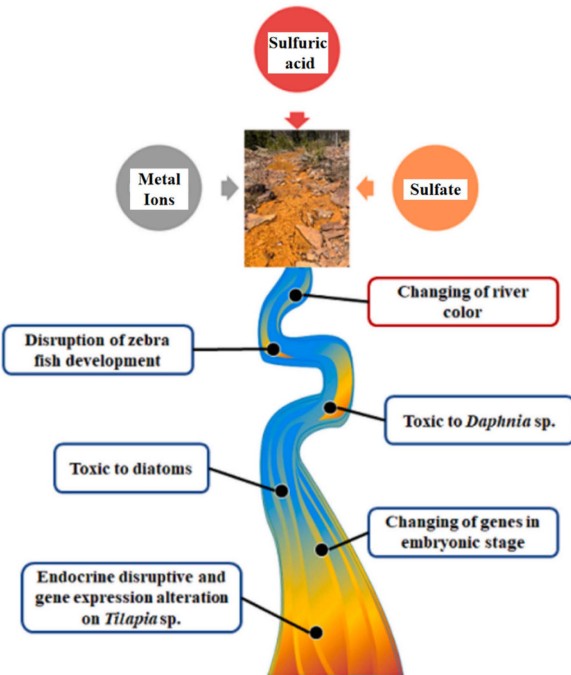

**Figure 3.** Summary of environmental effects of AMD. Adapted with permission from [50], copyrighter Elsevier, year 2022.

## 3. Resource Utilization of Acid Mine Drainage

With the rapid development of industry and the increasing demand for mineral resources, it has become extremely important to achieve resource recovery and reuse. Despite significant advances achieved in treating AMD, sustainability and costliness remain major challenges in the process. Thus, the exploration of sustainable treatment technologies has attracted the attention of researchers. At present, the possibility of recovering valuable resources from AMD has been illustrated by related scholars, and copper, iron, sulfuric acid, and reused fresh water have been successfully recovered from AMD at a laboratory scale.

### 3.1. Recovery of Water Resources from AMD

As industrialized production has brought enormous wealth to human beings, the damage to water resources is in an increasingly serious way. AMD is an important source of water resources, especially in semi/dry arid areas. Therefore, the recovery and reuse of water resources in AMD will provide easy access to the utilization of water resources.

Membrane separation technology has appeared as an important research topic for water recovery from AMD in recent years [60]. Wang et al. [61] demonstrated that wastewater recovery of 60% and flux reduction of 22% were achieved by applying the direct contact membrane distillation (DCMD) technique to pretreated AMD. Buzzi et al. [47] indicated that electrodialysis was suitable for recovering water from acid mine drainage with the desirable contaminant removal efficiencies (≥97%). However, the precipitation of iron at the surface of the cation-exchange membrane tends to cause a blockage of the membrane through the scaling phenomenon, which reduces the process efficiency. Masindi et al. [52] investigated the recovery of drinking water from acid mine drainage, and the drinking water could be prepared successfully with a reverse osmosis system. The pH of the drinking water obtained in this process was about 6.5 and the metal removal rate reached approximately 100%, meeting the drinking water quality as stipulated by SANS 241 standards.

Membrane separation technologies such as electrodialysis, membrane distillation, and permeation reaction are generally used to recover clean water from AMD. Among these methods, the permeation reaction has an advantage due to the fact that it can be used for the production of high-quality drinking water and reusable water. It reduces the utilization of fresh water and can meet the water needs of industry, agriculture, and ecosystems. Although membrane separation technology is beneficial to producing high-quality return water to meet the demand for clean production, there are some drawbacks associated with the use of the process, such as high energy demand and low water permeability, serious membrane fouling and low working period, and high operating costs. Accordingly, the process of recovering water from AMD is complex, and various processes have different strengths and weaknesses.

### 3.2. Recovery of Sulfuric Acid

Sulfuric acid is the most commonly applied and cheapest acid and can be widely used in chemical, wet metallurgy, and mineral processing, as well as in agriculture and pharmaceuticals [23]. For example, sulfuric acid plays a vital role in the production of phosphate fertilizers from wet phosphoric acid. Similarly, large amounts of sulfuric acid are also consumed in acid leaching and electrolytic deposition [23]. In the field of mineral flotation, sulfuric acid is commonly used as a regulator to improve the flotation performance of minerals [62,63]. As mentioned earlier, plenty of $H^+$ and $SO_4^{2+}$ ions are presented in the AMD, the recovery of sulfuric acid from AMD is consequently of increasing interest to researchers, and facilitates the resource utilization of acid mine drainage.

In previous studies, Nleya et al. [23] reported various process technologies that can recover sulfuric acid from AMD, including distillation, membrane separation techniques, solvent extraction, crystallization, and acid retardation techniques. Martí-Calatayud et al. [64] employed a three-compartment electrodialysis (ED) approach to recover sulfuric acid from AMD and the results demonstrated that a three-compartment ED cell consisting of AEM and CEM was able to obtain high purity sulfuric acid without the addition of additional chemical reagents, meeting the requirements of clean production. Cifuentes et al. [65] also confirmed that sulfuric acid could be effectively recovered from copper-containing electrolytes at a laboratory scale using electrodialysis. Kesieme et al. [66] used the direct contact membrane distillation (DCMD) process to recover sulfuric acid from acid mine drainage generated from mining operations. The experimental results showed that the sulfate separation efficiency could reach more than 99.9% when the temperature of the feed solution was maintained at 60 °C and the flux was kept in the range of 20–31 kg/m$^2$/h. In addition, Menzel et al. [67] successfully recovered sulfuric acid from AMD by electrodialysis and the purity of the recovered sulfuric acid was high enough to meet industrial demand.

However, the separation of positive and negative ions in ED systems is based on the fixed charge of the membrane by the ion exchange membrane. Therefore, the phenomenon of concentration polarization is accompanied by the membrane mass transfer process. The development of concentration gradients limited the mass transfer through the membrane. In addition, the presence of electrode reactions or membrane fouling can also affect the

performance of AMD treatment. The DCMD is capable of recovering high-quality sulfuric acid from the waste acid solution, but its application is limited in practice due to its high production cost and process complexity.

### 3.3. Subsection Recovery of Valuable Metals and Synthetics

### 3.3.1. Recovery of Dissolved Metal Ions from AMD

AMD contains large amounts of dissolved valuable metal ions such as Fe, Cu, Zn, Pb, Al, Mn, et al., which will pose a potential threat to human health and the surrounding environment. The recovery of these valuable metal ions can solve the potential pollution issues as well as realize the reuse of valuable resources. In recent years, the research focuses have shifted gradually from AMD prevention and treatment to the recovery of valuable metal ions. Membrane distillation [68], nanofiltration [69], chemical oxidation techniques [70], electrochemical [71], selective precipitation [57,72], and solvent extraction [69] have been widely applied to recover metal ions from AMD. For instance, Chen et al. [73] adopted fractional precipitation to recover Fe, Cu, Zn, and Mn from AMD, and the precipitation sludge was roasted to obtain iron oxide red with $Fe_2O_3$ content of 85.18%. Meanwhile, the flotation process was performed to obtain Cu and Zn rough concentrates with recoveries of 72.66% and 76.18%, respectively. Brewster et al. [54] employed a staged electrochemical neutralization process to treat AMD and selectively recover metals and rare earth elements from the precipitation.

Table 2 lists the processes for the recovery of valuable metals from AMD. From these data, it can be concluded that the typical metal ions, including Fe, Cu, Zn, Al, and Mn could be successfully recovered from AMD. Among the many treatment methods, selective precipitation is still the most common method to recover metals from AMD due to its low operating cost and high efficiency; however, the tendency to produce sludge with high water content during the treatment process is its main drawback. Also, the electrochemical and membrane technologies are also used to effectively recover metals from AMD, but the complicated process and high operating costs of these treatment technologies render them difficult to apply at real mine sites and are mostly performed at a laboratory scale. Consequently, the feasibility of recovering metals from AMD must be evaluated according to the economic value of the metals, the concentration of the metals in AMD, and the extraction efficiency.

**Table 2.** Techniques for selective recovery of metal ions from AMD.

| Recovery Methods | Metal Ions Recovery | Description | References |
|---|---|---|---|
| Membrane Distillation and Adsorption System | Cu | The valuable copper resources were selectively recovered from the synthesized AMD solution by membrane distillation and adsorption systems. The pH of AMD was adjusted to the range of 5.0–5.2, and the KOH-treated AMD was continued to concentrate the solution by the direct contact membrane distillation (DCMD) process with the aim of achieving the selective adsorption of Cu for the multi-modified mesoporous silica SBA-15 material and the amount of Cu adsorption reached 24.53 mg/g. | [68] |
| Metal Sulfide Precipitation and Membrane Filtration Process | Cu | A novel process combining metal sulfide precipitation and membrane microfiltration was investigated for the recovery of copper from synthetic AMD. The recovery of copper was close to 100% and turbidity values in the treated solution were lower than 2 NTU for sulfide stoichiometric dosages of 120%. | [67] |

**Table 2.** *Cont.*

| Recovery Methods | Metal Ions Recovery | Description | References |
|---|---|---|---|
| Nanofiltration (NF) and Solvent Extraction (SX) | Cu | The feasibility of recovering copper from actual AMD was investigated by pilot-scale tests. Nanofiltration (NF) was applied to concentrate copper from AMD solutions, and then solvent extraction (SX) was performed to extract copper from the concentrated solution of NF. The results showed that the combination of NF and SX is promising for copper recovery and the copper recovery rate reached 97%. | [74] |
| Selective Precipitation (SP) | Cu and Zn | A field trial was conducted for the recovery of Cu and Zn from AMD produced at an operating mine using a selective precipitation (SP) pilot plant. Under the AMD condition of 1.4 L/min, Cu and Zn precipitates with a purity of 80% and a precipitation rate of 90% can be obtained. | [75] |
| Chemical Oxidation Technology ($H_2O_2$-NaOH Technology) | Fe | The chemical oxidation pilot process was used to treat AMD with high concentration of Fe. Under acidic conditions, $H_2O_2$ was able to rapidly oxidize $Fe^{2+}$ to $Fe^{3+}$. Followed by the addition of NaOH to adjust the pH to 3.8, Fe-contained precipitates were formed and the average Fe content in the recovered sludge was 26.85%. | [70] |
| Electrochemical Reactions | Fe, Al, Cu, Zn, and Ni | Electrochemical reactions are possible to oxidize Fe(II) to Fe(III) while producing neutralizing agents (containing a high concentration of hydroxide) for the selective recovery of dissolved metals (Fe, Al, Cu, Zn, and Ni) from AMD. | [71] |
| Sequential Selective Precipitation and Fluidized Bed Homogeneous Crystallization (FBHC) | Fe and Al | The $Fe^{2+}$ and $Al^{3+}$ were recovered from actual AMD using a combination of sequential selective precipitation and fluidized bed homogeneous crystallization (FBHC). Under the conditions of pH of about 9.25, $[H_2O_2]/[Al(III)]$ molar ratio of 2.0 and upward flow rate (U) of 30.5 m/h. Ferric hydroxide ($Fe(OH)_3$) and bayerite ($\alpha$-$Al(OH)_3$) pellets were prepared sequentially. | [76] |
| Sequential Selective Precipitation | Fe, Al, and Mn | The selective precipitation of dissolved iron, aluminum, and manganese in the AMD from the Samma-Taejeong coal mine by adding oxidants and neutralizers was investigated. In the case of oxidation ($H_2O_2$) and then neutralization of AMD, the sequence of metal removal was Fe>Al>Mn in order, and the recovery of dissolved Fe, Al, and Mn reached 99.2–99.3%, 70.4–82.2%, and 37.8–87.5%, respectively. | [57] |

### 3.3.2. Synthesis and Application of Valuable Minerals

Researchers have proposed new methods to treat AMD by synthesizing valuable minerals. The investigation indicated that the valuable elements obtained from AMD can be used as catalysts, adsorbents, and mineral materials [77–79]. Akinwekomi et al. investigated the beneficiation of acid mine drainage (AMD), and the results indicated that goethite, hematite, and magnetite with high purity (100%) could be synthesized from AMD. Additionally, gypsum was synthesized, and drinking water was also reclaimed. This study will foster the concept of circular economy and waste beneficiating, thus curtailing the impacts of AMD on different spheres of the environment [77,80]. Iron nanoparticles have been identified as effective adsorbents and catalysts for various environmental pollutants attributed to their large specific surface area and high surface activity [81]. Cheng et al. [82] adopted fuel cell technology to treat AMD and produced spherical iron oxide nanoparticles. This can be turned into goethite (a-FeOOH) after drying. Flores et al. [28] also investigated the recovery of iron oxides from acid mine drainage and their application as adsorbents or catalysts. It is worth noting that these iron oxide particles can be used in pigments and other products, thus allowing the process to be sustainable.

The synthesis of valuable minerals from AMD (e.g., iron oxide nanoparticles using Fe ions originating from AMD) and applying them as adsorbents or catalysts for pollutants is a worthy method to be advocated for AMD treatment. However, the synthesis of the majority of iron oxide nanoparticles occurred at high pH and specific temperatures (100–700 °C) [28], which undoubtedly increased the complexity of the process as well as the treatment cost. Moreover, it is difficult to apply one treatment method to the total AMD treatment owing to the wide range of sources, the high variability of wastewater properties, and the complexity of the components, which results in different treatment methods required for AMD in different locations. Furthermore, most of the methods mentioned above regarding the synthesis of valuable minerals from AMD mainly focus on the treatment of synthetic AMD, whereas the treatment capacity of these technologies in raw AMD is yet to be explored in detail.

## 4. Application of AMD in Fenton Oxidation Technology

Fenton oxidation, one of the advanced oxidation processes (AOPs), is currently the most promising technology for wastewater treatment [83,84]. The reaction mechanism is based on the reaction between ferrous ions ($Fe^{2+}$) and hydrogen peroxide ($H_2O_2$) under acidic conditions to generate hydroxyl radicals ($\bullet OH$), which plays a crucial role in the Fenton reaction [85–87]. The generated $\bullet OH$ possesses a high oxidation potential of 2.80 V, which has a strong oxidative capacity to effectively degrade organic pollutants in wastewater [86,88]. The reaction equation (Equations (1)–(7)) is as follows [87]:

$$Fe^{2+} - H_2O_2 \rightarrow OH^- - \bullet OH - Fe^{3+} \tag{1}$$

$$OH\bullet - Fe^{2+} \rightarrow OH^- - Fe^{3+} \tag{2}$$

$$Fe^{3+} - H_2O_2 \rightarrow Fe - OOH^{2+} - H^+ \tag{3}$$

$$Fe - OOH^{2+} \rightarrow HO_2\bullet - Fe^{2+} \tag{4}$$

$$Fe^{2+} - HO_2\bullet \rightarrow Fe^{3+} - HO_2^- \tag{5}$$

$$Fe^{3+} - HO_2\bullet \rightarrow Fe^{2+} - H^+ - O_2 \tag{6}$$

$$OH\bullet - H_2O_2 \rightarrow H_2O - HO_2\bullet \tag{7}$$

However, the conventional Fenton oxidation process produces a large amount of iron sludge, which has the risk of secondary pollution [89–91]. Moreover, the reaction is usually performed under a strongly acidic medium environment, which requires additional acid to adjust the pH, and also requires the addition of ferrous ions, increasing the cost of chemical reagents. The Fenton reaction process consists of four main stages: pH adjustment, oxidation reaction, neutralization coagulation, and precipitation [92]. Among these stages, the key oxidation phase of $H_2O_2$ decomposition requires the presence of $H^+$ and $Fe^{2+}$ to promote $OH\bullet$ production, as well as to improve the $H_2O_2$ utilization under an acidic environment. AMD is commonly acidic and contains $Fe^{2+}$, which makes it suitable for the fabrication of Fenton catalytic reagents.

Sun et al. [93] investigated a new air-cathode fuel cell installation for the in situ fabrication of heterogeneous electro-Fenton catalysts using ferrous iron in AMD. Three different types of nanostructured iron oxide/graphite felt composites, FeOOH/GF, $Fe_2O_3$/GF, and $Fe_3O_4$/GF, were fabricated from the synthesized AMD. The results showed that with Rhodamine B (Rh B) as the test pollutant, the removal rates of Rh B of the three composites under neutral pH conditions are 62.5%, 95.4%, and 95.6%, respectively. Sun et al. [94] also employed air-cathode fuel cell technology (AC-FC) for the in situ fabrication of iron oxide/carbon composites with electro-Fenton catalytic activity using $Fe^{2+}$ originating from AMD. In addition, the effect of coexisting metal cations in AMD excluding $Fe^{2+}$ on the fabrication process of electro-Fenton catalysts was also investigated. The AC-FC was then employed to degrade organic pollutants in the actual AMD. The test results indicated that the AC-FC exhibited favorable performance in treating the actual AMD, and the

treated wastewater reached the specified discharge standards. Meanwhile, $Fe^{2+}$ ions were recovered from the actual AMD as a material source for the heterogeneous electro-Fenton catalyst, which enabled the effective utilization of valuable resources in AMD, and was in line with the concept of green sustainable development.

Huang et al. [95] investigated the $Fe^{2+}$ and $Mn^{2+}$ catalysts on the performance of electro-Fenton (EF) degradation of antibiotic ciprofloxacin (CIP). By using acid mine drainage (AMD) rich in iron and manganese to replace the homogeneous solution in EF, the CIP removal efficiency of 89.00% in 60 min was achieved under optimal conditions and might assign new perspectives for organic pollutant removal by utilizing AMD. Mahiroglu et al. [92] investigated the treatment of combined acid mine drainage (AMD)-flotation circuit effluents from the copper mine via Fenton's process. The results suggested that the utilization of existing pH and $Fe^{2+}$ stemmed from AMD, low $H_2O_2$ requirements, and up to 98% treatment performances in COD, turbidity, color, $Cu^{2+}$, and $Zn^{2+}$ made the proposed treatment system promising.

The Fenton reagent is prepared to degrade organic pollutants in wastewater by utilizing the acidity and $Fe^{2+}$ content of AMD itself, which will considerably economize the chemical reagent cost of oxidation reaction and can effectively treat wastewater. These pollutants are eventually mineralized into $CO_2$, $H_2O$, and inorganic ions, which completely degrade them and achieve the purpose of treating waste with waste. Consequently, it is a worthy method to be advocated for the preparation of Fenton oxidation catalysts using AMD which contain acid and a high concentration of $Fe^{2+}$ ions.

## 5. Application of AMD in Beneficiation

Currently, AMD treatment technologies have been reported in detail by previous studies and are widely used in the majority of mines, however, most of them are mainly concentrated on the removal of acidity, metal, and sulfate content of the AMD itself. These available remediation approaches usually require the continuous supply of chemicals and energy, expensive maintenance, complex processes, and the long-term monitoring of affected ecosystems [96,97]. Most of them are both unaffordable and unsustainable. Indeed, assiduous efforts on the recovery of valuable AMD constituents have been ongoing, albeit with minimal success. The valuable resources in AMD fail to be recycled desirably. Thus, it is extremely urgent to exploit the economically sustainable technologies for AMD treatment. In recent years, the comprehensive utilization of AMD in the beneficiation field has attracted soaring interest. This is rooted in the fact that major elements ($H^+$, $Fe^{2+}$, and non-ferrous metal ions) in AMD render this natural acidic wastewater a promising regulator.

Zhou [98] used acidic wastewater mixed with sodium sulfide for the activation of lime-depressed pyrite, and practical production showed that the grade of sulfur concentrate in Wushan copper ore in China was improved by about 4% from the original 32.45%, and the recovery was increased by approximately 9.15% from the original 42.85%, and the index of sulfur separation operation was remarkably stable. Bai et al. [35] investigated the activation mechanism of lime-depressed pyrite flotation by adding AMD with the aim of replacing the conventional activators (for instance sulfuric acid, oxalic acid, and copper sulfate) and gearing towards a cleaner production concept of copper sulfide ore. The corresponding mechanism diagram was shown in Figure 4. The results of pure mineral flotation tests showed that the flotation recovery of pyrite inhibited by lime was increased by 64% when the volume ratio of AMD to HAS (high alkali system) was 3:1. Actual ore tests further confirmed that AMD could effectively activate the pyrite flotation with the SBX collector. Consequently, the application of AMD in pyrite beneficiation can not only reduce the consumption of chemical reagents but also provide a new method for the sustainable treatment of AMD.

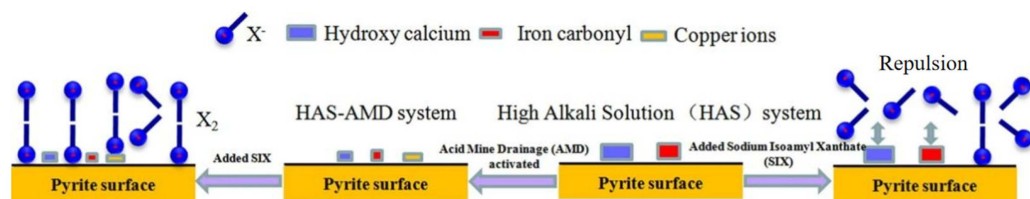

**Figure 4.** Adsorption model diagram of the HAS-depressed pyrite flotation with AMD addition.

In addition, we have investigated an innovative process for the flotation of chalcopyrite activated by adding AMD [99]. The results revealed that AMD promoted the desorption of hydrophilic calcium species on the chalcopyrite surface, exposing the fresh surface of chalcopyrite, and increasing the adsorption amount of collector onto chalcopyrite surfaces. This research provides an innovative option for the comprehensive utilization of AMD and the recovery of low-grade chalcopyrite from tailings. The application of AMD to beneficiation has vital practical significance for the emission reduction in AMD and the increase in beneficiation profits.

AMD can also be used as a regulator of flotation pulp. Gong et al. [100] applied acidic wastewater directly to iron sulfide ore beneficiation and adjusted the slurry to a suitable pH value, which achieved the qualified sulfur concentrate to meet the production of the sulfuric acid industry. Duan [101] mixed acidic wastewater with the tailing slurry and applied the supernatant to the grinding and flotation production. The results indicated that the acidic wastewater could replace the fresh water and a desirable flotation index could be obtained. In addition, Zhan [102] performed leaching treatment of copper-bearing waste rock from the Dexing copper mine in china using AMD as a leaching agent and acquired relatively good leaching results after three cycles with the help of bacterial oxidation. It consequently not only solved the environmental pollution issue of AMD but also recycled the copper metal in the waste rock.

## 6. Conclusions and Future Perspectives

AMD is an acidic solution containing high concentrations of dissolved metals and sulfates, which has serious impacts on the environment and human health due to its complex composition and high emissions. During the last few decades, the prevention and treatment of AMD have been extensively reported, and the specific choice of treatment method always depends on the composition of the wastewater, the concentration of metal ions, and the specific geographical environment. Thus, the achievement of sustainable treatment of AMD still faces extreme challenges.

In view of the properties of AMD, growing interests are the recovery of sulfuric acid, water resources, and valuable constituents from AMD; plenty of sustainable treatment options have been evaluated. Currently, selective precipitation, chemical neutralization, membrane separation processes, ion exchange, chemical oxidation techniques, electrochemistry, and fuel cell technologies have been widely studied for the recovery and reuse of valuable resources from AMD with promising applications. Despite the positive outcomes of the different types of treatment, they have associated technical issues, and many efforts must be paid to the industrial application of these technologies.

Fenton oxidation technology possesses the advantages of fast reaction, mild conditions, and simple equipment. The in situ fabrication of heterogeneous electro-Fenton catalysts using AMD with low pH and rich $Fe^{2+}$ not only reduces the reagent dosing for the Fenton oxidation reaction but also enables the recycling of valuable resources in AMD. However, the research on the reaction mechanism related to the fabrication of Fenton reagents using AMD as materials in the wastewater treatment process needs to receive further discussion. Future works should focus on the effect of various metal ions in AMD on the catalytic activity of the Fenton system and the appropriate expansion of the pH range of the reaction.

The application of AMD in beneficiation fields, such as the activation of pyrite and chalcopyrite flotation, provides easy access to the innovative utilization of AMD in copper

sulfide ore. This option can not only realize the efficient recovery of target minerals, increasing the beneficiation profits, but also solves AMD's negative impacts on the receiving environment at its source and facilitates the clean production of mines. Whereas scarce information is available regarding the application of AMD in beneficiation fields, the majority of the works are still in the laboratory stage. Notably, each approach has its technicalities, and it is recommended that more blended techniques be explored to realize the resource utilization of acid mine drainage.

**Author Contributions:** Conceptualization, J.Y. and J.L.; methodology, Y.B.; software, J.Y.; validation, S.B.; formal analysis, J.Y. and Z.D.; investigation, S.B.; resources, S.B.; data curation, J.Y. and S.B.; writing—original draft preparation, J.Y.; writing—review and editing, J.Y. and S.B.; visualization, S.B.; supervision, S.W.; project administration, J.Y. and S.B.; funding acquisition, S.B. All authors have read and agreed to the published version of the manuscript.

**Funding:** This work was financially supported by the National Natural Science Foundation of China (Grant No. 52164021) and the Natural Science Foundation of Yunnan Province (Grant No. 2019FB078).

**Institutional Review Board Statement:** Not applicable.

**Informed Consent Statement:** Not applicable.

**Data Availability Statement:** Not applicable.

**Acknowledgments:** We would appreciate the Regions of the National Natural Science Foundation of China (Grant No. 52164021) and the Natural Science Foundation of Yunnan Province (Grant No. 2019FB078) for providing financial supports for this research project.

**Conflicts of Interest:** The authors declare that they have no known competing financial interest or personal relationships that could have appeared to influence the work reported in this paper.

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
