# Peer review of "Resource Utilization of Acid Mine Drainage (AMD): A Review"

_water, doi:10.3390/w14152385_

Round 1

Reviewer 1 Report

This paper has assembled recent references on techniques to remediate AMD in order to obtain commercially valuable products with the potential to make remediation sustainable. Although the paper states that it intends to “identify the strengths and weaknesses of each treatment technology”, in fact it just uncritically summarises each paper. While it is useful to have a compilation of recent papers on this topic, this review would be much more useful if it actually identified the strengths and weaknesses of each treatment, and those that have the greatest potential. In the conclusions there is an attempt to assess the effectiveness of the different processes, but this could be considerably more detailed. In particular, it is notable that virtually all the papers cited are laboratory studies; few, if any of these have been translated into viable treatments at AMD sites. This is largely because they are almost all too expensive or impractical to apply at a field scale.

The review also suffers from a rather idiosyncratic organisation. Some papers are summarised in a table, some are summarised in the text under specific headings, others are summarised under general headings.

This paper could become a standard reference on the topic if it was more logically organised and critically evaluated the different technologies in terms of expense and scalability.

Specific comments

1.       The paragraph on the health effects of AMD (lines 119-131) is unnecessary, as is Fig. 3.

2.     Table 1 shows a very selective approach to the data; it includes country-wide averages as well as analyses from specific mines. If it is intended to demonstrate the variability of AMD composition, analyses from 5-10 individual AMD sites would be sufficient. In any case, the relevance of this table in this paper is unclear – it could easily be deleted.

3.       The studies covered in lines 142-151 belong under one of the headings in the following part of the paper.

4.       How did Masindi et al avoid membrane fouling?

5.     Table 2 occupies more space than describing these studies in the text. The headings under which the studies are described need to be incorporated into the headings in the following text. This will avoid the present situation where sequential precipitation is split into several different lines of the table.

6.     Techniques that produce Fe and Al precipitates are unlikely to be economic, because the value of these minerals is so low. Describing such precipitates as “valuable elements” (line 213) is very misleading.

7.     The description of the use of AMD in flotation (lines 285-304) is much more detailed than for any other technique. This seems to be because the authors of this review wrote some of these papers. This section could be easily condensed; the Bai et al study is described in too much detail.

8.     The discussion of many of the processes seems to assume that all AMD will behave in the same way. From the variability evident in Table 1, this is very unlikely, and could be particularly relevant to the use of AMD in flotation.

Minor points

Line 100 – the oxidation occurs in water exposed to air

Line 102 – delete ‘eventually’

Lines 103 and 112 – wrong use of ‘predominates’

There are a few spelling/grammatical mistakes, e.g. ‘essay’ line 352, ‘have’ line 138.

Author Response

Thank you for the affirmation to our study and precious suggestion for this manuscript. We carefully and thoroughly revise our manuscript according to your comments and suggestions. Thank you for your careful review. We have added the strengths and weaknesses of each treatment in the corresponding position in the original manuscript, as well as an evaluation of those treatments that have the greatest potential in terms of expense and scalability. Moreover, we do our best to polish the logical organization for this manuscript. Specially, the papers summarized in a table are further condensed with the usual-accepted style according to other reviewer comments. While the papers are summarized in the manuscript and others are under specific headings to present the expression of the first summary and then divided points. All of changes are marked in red and listed in the change list. We hope you would find the revised manuscript satisfactory.

Reviewer 2 Report

Extensive editing to correct English syntax is needed.

The technical review is pedestrian.

In at least one case, evidence is drawn from an experiment using a synthetic AMD solution.  That generally leads overly optimistic interpretations of treatment performance.

The literature review is inadequate.

Author Response

Thank you for your affirmation to our research and precious suggestion. We are grateful for your earnest help to improve our manuscript. We have tried our best to carefully and comprehensively examine and correct the entire manuscript, including the writing and language, to make it more readable. All of changes are marked in red and listed in the change list.  We hope you would find the revised manuscript satisfactory.

Round 2

Reviewer 1 Report

The revisions are fine, except they have introduced a number of grammatical errors.